# Bile Acids and Bilirubin Role in Oxidative Stress and Inflammation in Cardiovascular Diseases

**DOI:** 10.3390/diseases12050103

**Published:** 2024-05-14

**Authors:** Angela Punzo, Alessia Silla, Federica Fogacci, Matteo Perillo, Arrigo F. G. Cicero, Cristiana Caliceti

**Affiliations:** 1Department of Biomedical and Neuromotor Sciences, Alma Mater Studiorum, University of Bologna, 40126 Bologna, Italy; angela.punzo2@unibo.it (A.P.); matteo.perillo2@unibo.it (M.P.); cristiana.caliceti@unibo.it (C.C.); 2Biostructures and Biosystems National Institute (INBB), 00136 Rome, Italy; 3Department for Life Quality Studies, Alma Mater Studiorum, University of Bologna, 47921 Rimini, Italy; alessia.silla2@unibo.it; 4Hypertension and Cardiovascular Risk Research Center, Medical and Surgery Sciences Dept., Alma Mater Studiorum, University of Bologna, 40138 Bologna, Italy; federica.fogacci@studio.unibo.it; 5Cardiovascular Medicine Unit, IRCCS AOU di Bologna, 40138 Bologna, Italy; 6Interdepartmental Centre for Industrial Agrofood Research—CIRI Agrofood, University of Bologna, 47521 Cesena, Italy

**Keywords:** bile acids, bilirubin, cardiovascular disease, oxidative stress, inflammation

## Abstract

Bile acids (BAs) and bilirubin, primarily known for their role in lipid metabolism and as heme catabolite, respectively, have been found to have diverse effects on various physiological processes, including oxidative stress and inflammation. Indeed, accumulating evidence showed that the interplay between BAs and bilirubin in these processes involves intricate regulatory mechanisms mediated by specific receptors and signaling pathways under certain conditions and in specific contexts. Oxidative stress plays a significant role in the development and progression of cardiovascular diseases (CVDs) due to its role in inflammation, endothelial dysfunction, hypertension, and other risk factors. In the cardiovascular (CV) system, recent studies have suggested that BAs and bilirubin have some opposite effects related to oxidative and inflammatory mechanisms, but this area of research is still under investigation. This review aims to introduce BAs and bilirubin from a biochemical and physiological point of view, emphasizing their potential protective or detrimental effects on CVDs. Moreover, clinical studies that have assessed the association between BAs/bilirubin and CVD were examined in depth to better interpret the possible link between them.

## 1. Introduction

Despite great scientific progress in recent years, cardiovascular diseases (CVDs), including heart disease, stroke, and atherosclerosis, are still the leading causes of morbidity and mortality globally [1]. The prevalence of these diseases is influenced by factors such as lifestyle (especially diet and physical activity) and access to healthcare. Recent research has focused on understanding the underlying mechanisms of CVDs, identifying risk factors, and developing innovative treatments.

Advances in areas such as genetics, personalized medicine, and interventions to address risk factors have certainly contributed to improvements in cardiovascular (CV) care [2].

Oxidative stress and inflammation are processes implicated in the pathogenesis of various CVDs, including atherosclerosis, hypertension, myocardial infarction, and heart failure [3]. Oxidative stress is a condition characterized by an imbalance between the production of reactive oxygen species (ROS) and the body’s ability to detoxify them or repair the resulting damage. ROS, including free radicals like superoxide anion and hydroxyl radical, are molecules with unpaired electrons that can damage cellular structures, including proteins, lipids, and DNA [4]. On the other hand, inflammation is characterized by increased blood flow, vascular permeability, and the production of pro-inflammatory cytokines, chemokines, and acute-phase proteins. While acute inflammation is a protective response aimed at removing harmful stimuli and promoting tissue repair, chronic inflammation can contribute to the pathogenesis of several diseases, including CVDs, diabetes, and cardiometabolic disorders [5].

While acute inflammation is a protective response aimed at removing harmful stimuli and promoting tissue repair, chronic inflammation can contribute to the pathogenesis of several diseases, including CVDs, diabetes, and cardiometabolic disorders [5].

Oxidative stress and inflammation can mutually influence and amplify each other. ROS can act as signaling molecules to activate various inflammatory pathways, such as nuclear factor-kappa B (NF-κB) and mitogen-activated protein kinases (MAPKs), leading to the production of pro-inflammatory cytokines and chemokines. In turn, inflammatory mediators can induce the production of ROS by activating NADPH oxidases and other ROS-generating enzymes in immune and endothelial cells. This creates a feed-forward loop wherein oxidative stress promotes inflammation, and inflammation exacerbates oxidative stress, leading to tissue damage and dysfunction [6]. BAs are primarily known for their role in the digestion and absorption of dietary fats in the gastrointestinal tract; however, their role in CVD development is still controversial. A recent study has suggested that the most hydrophilic BAs (i.e., ursodeoxycholic acid (UDCA)) may help in CVD prevention by acting as a signaling molecule [7], while previous research has shown that other BAs (i.e., the lithocholic acid (LCA), which is the most hydrophobic secondary BA) can be involved in oxidant and inflammatory injury. Thus, the hydrophobicity of individual BAs in the serum bile acid pool may be of relevance [8]. Bilirubin is a yellow pigment formed during heme breakdown, a component of hemoglobin in red blood cells, and stored in the gallbladder. Bilirubin possesses antioxidant and anti-inflammatory properties; indeed, it can scavenge and neutralize free radicals [9]. Increased serum concentration of bilirubin (hyperbilirubinemia > 1.5 mg/dL) is typical of pathologic conditions, such as jaundice or Gilbert Syndrome (GS), and often indicates liver diseases. However, bilirubin concentration at the upper end of the physiological range was found to be associated with a reduced risk of coronary artery disease, peripheral artery disease, and other manifestations of CVDs; this is believed to be due to bilirubin’s anti-inflammatory and antioxidant properties [10]. Clinical studies are still ongoing to determine whether BAs and bilirubin can be considered predictive biomarkers for CVDs, and results are sometimes controversial.

This review appraises, synthesizes, and discusses as many as possible preclinical and clinical studies that have assessed the relation between BAs and bilirubin with CVDs, providing a comprehensive overview of this topic that, to the best of our knowledge, has never been fully investigated before.

## 2. Literature Search and Screening

### 2.1. Search and Screening Strategy (Methods)

A systematic search of studies published between 1972 and 2024 on the relationship between BAs and bilirubin with CVDs was performed. The literature of interest was searched in PubMed, Scopus, and Web of Science (Core collection). The queries were designed to retrieve all papers published in peer-reviewed journals and containing in the titles and/or in the abstract the words “bile acids” or “bilirubin” and the words “cardiovascular disease” or “cardiovascular system”. The search was limited only to papers available in English.

After merging and deduplicating the results of the search, we performed a two-phase screening procedure to identify the papers coherent with the research question. To be considered relevant, the papers had to comply with the following inclusion criteria:-The paper discusses BAs and bilirubin synthesis and metabolism, physicochemical properties and their receptors and signaling pathways;-The paper includes findings from human and animal studies (if relevant) with the support of preclinical data related to the role of BAs and bilirubin in CVDs.

The non-availability of the full text was a criterion for exclusion.

In the first phase, two researchers screened the titles and abstracts separately, consulting each other in case of doubt regarding the selection of specific studies related to the link between oxidative stress, inflammation, and CVDs. In the second phase, a third operator assessed the full text of the papers considered potentially relevant during the first phase.

The first phase of screening was performed using the software ASReview (v1.5) [11], a tool that utilizes active learning and natural language processing (NLP) to study the features of the articles conforming with the inclusion criteria of a review. This allows tthe screening process to be sped up, as the tool iteratively proposes to the reviewer the next article to screen based on the features learned [12]. Therefore, most of the papers of interest are found by the reviewers in the initial part of the abstract screening. In this case, following the example of Perillo et al. [13] a two-fold stopping rule for this phase of screening was established: screening was stopped if more than 80 irrelevant papers in a batch of 100 papers were found (stopping rule #1) or when the screeners had performed 8 h of screening each (stopping rule #2), whichever happened first.

### 2.2. Search and Screening Results

The results of the search and screening activities are summarized in the PRISMA flowchart in Figure 1.

Performing the search described in the previous section, we found 8504 potentially relevant papers on PubMed, 6001 on Scopus, and 3084 on Web of Science. After deduplication, we were left with 14,841 potentially relevant unique papers.

Such papers were screened with ASReview until less than 20 relevant papers in a batch of 100 papers were found (stopping rule #1). The stopping rule was fulfilled after screening 941 papers, 381 of which were considered potentially relevant and 560 of which were considered irrelevant. A total of 13,900 records were not screened at all since the stopping rule was reached before, so it is reasonable to expect that none or very few of them were potentially relevant.

After screening the full text of the 381 papers considered potentially relevant, we were left with 104 relevant papers, which entered the review.

## 3. Bile Acids: Biochemistry and Physiology

BAs are the end products of cholesterol metabolism, which is tightly regulated by the liver. BAs are detergent-like amphipathic molecules representing the major constituents of bile, a hepatic aqueous secretion consisting of bile salts, cholesterol, phospholipids, and bilirubin, which is fundamental for the intestinal absorption of nutrients such as dietary fats, steroids, drugs, liposoluble vitamins, and other lipophilic compounds [14]. In recent years, BAs have emerged not only for their function in emulsification and intestinal absorption of dietary lipids but also as signaling molecules involved in regulating hepatic lipid levels, energy, and glucose homeostasis [15].

BA biosynthesis occurs through two main processes: the classical pathway, which occurs in the liver, and the alternative/acidic pathways, which can also involve macrophages, adrenal glands, and the brain [16]. In humans, primary BA cholic (CA) and chenodeoxycholic acid (CDCA) are exclusively synthesized from cholesterol in pericentral hepatocytes [17] through multiple enzymatic steps, whose rate-limiting one is the cholesterol 7a-hydroxylase (CYP7A1) that ultimately converts cholesterol intermediates into primary BAs (Figure 2) [16]. Once produced, primary BAs are conjugated with glycine or taurine amino acids to increase their solubility and reduce cytotoxicity [18]. Then, they are actively secreted into the bile canalicular lumen and subsequently stored in the gallbladder [19] as bile salts within mixed micelles made of phospholipids and cholesterol, facilitating the excretion of cholesterol and other lipophilic molecules [20]. During food intake, the cholecystokinin hormone (CCK) stimulates bile release from the gallbladder into the duodenum lumen to facilitate the absorption of dietary lipids and fat-soluble vitamins [21]. In the terminal ileum, 95% of conjugated BAs are reabsorbed through active transport via the apical sodium-dependent BA transporter (ASBT) [22] and reach the liver through the enterohepatic circulation (EHC).

The small fraction of BAs that are not reabsorbed and remain in the large intestine (~5%) undergoes different gut bacteria reactions (e.g., deconjugation, desulfation, dehydrogenation, dehydroxylation, and epimerization) [23]. Bile salt hydrolase (BSH) enzymes present in the intestinal microbiota of some bacteria species, like Clostridium, Lactobacillus, Listeria, and Enterococcus, mediate BA deconjugation, thus releasing free BAs and amino acids moieties [24]. Subsequently, colonic anaerobic bacteria of the genera *Bacteroides*, *Clostridium*, *Eubacterium*, *Lactobacillus*, and *Escherichia* mediate the dehydroxylation at the carbon-7 position in deconjugated primary BAs, CA, and CDCA, to generate the corresponding secondary BAs: deoxycholic (DCA) and LCA [25]. UDCA is another human secondary BA that is produced in small quantities (less than 1%) by gut bacteria following epimerization of the 7α-hydroxyl group of deconjugated CDCA [26]. Secondary BAs can be passively absorbed from the large bowel and transported to the liver or lost in the fecal output [27].

As previously mentioned, the role of BAs as signaling molecules has recently emerged due to their ability to selectively bind nuclear and cell surface receptors, which in turn activate several pathways [28]. Among these receptors, two of the best characterized are the farnesoid X receptor (FXR), recognized as the nuclear receptor [29], and the Takeda G protein-coupled receptor 5 (TGR5), a G-protein coupled receptor [30]. Both receptors are selectively activated by BAs. Other nuclear receptors are the Pregnane-X-receptor (PXR) [31], the Constitutive Androstane Receptor (CAR) [32], the Vitamin D Receptor (VDR) [33], the sphingosine-1-phosphate receptor (S1PR) [34], and other G-protein coupled receptors, able to bind BAs in a non-exclusive manner [35].

FXR is highly expressed in the liver, kidney, and intestine but also in the coronary arteries, aorta, atherosclerotic arteries, and cardiomyocytes in the CV system [14]. It regulates key genes involved in the metabolic process of BA synthesis, transport, reabsorption, and metabolism of carbohydrates and lipids, and it is also involved in the modulation of vascular inflammation [36].

TGR5 is expressed in various cell types, including gallbladder epithelial cells (cholangiocytes), gallbladder smooth muscle cells, Kupffer cells, intestinal L cells, pancreatic β cells, aortic endothelial cells, cardiomyocytes, skeletal muscle cells, nerve cells, and brown adipocytes [37]. Moreover, it is involved in the regulation of different cell signaling pathways, BA homeostasis, glucose metabolism, monocyte adhesion, and inflammatory response modulation [38].

## 4. Bilirubin: Biochemistry and Physiology

Another important bile constituent is bilirubin, which is the catabolic product of the heme pathway, released after red blood cell (RBC) lysis (Figure 2) [39]. During erythrocyte apoptosis, the globin and the heme ring are released [40] by the reticuloendothelial system. Next, the heme ring is opened thanks to the activity of the heme oxygenase (HO)-1, which releases iron, carbon monoxide, and biliverdin. Subsequently, biliverdin is reduced by biliverdin reductase (BVR) into unconjugated bilirubin (UCB), which is transported in blood tightly bound to human serum albumin (HSA) before its uptake by the hepatocytes. Only less than 0.1% of UCB is unbound to albumin (the so-called free bilirubin); this fraction determines the biological activities of bilirubin. In physiologic conditions, once it reaches the liver, 99% of bilirubin is conjugated by UDP-glucuronyltransferase (UGT) to glucuronic acid, a water-soluble form that allows its secretion into the bile [41].

After conjugation into the liver and bile secretion, bilirubin is released postprandially with BAs as a bile constituent, and once in the gut, is reduced by gut bacteria to bilirubin metabolic compounds, also called urobilinoids [42]. Gut bacteria of the genera *Clostridium* and *Bacteroides* are known to be able to deconjugate bilirubin through β-glucuronidase enzyme, as well as to reduce bilirubin to urobilinoids like urobilinogen, urobilin, and stercobilinogen, which is then oxidized into stercobilin [43]. UCB and its derivatives are reabsorbed both in the small and large intestine enterocytes from the ileum to the colon. Subsequently, they are released into the portal circulation and delivered to the liver, where they are recycled into the biliary flow [44]. Moreover, small quantities of urobilinogen are released into feces as pigments or reabsorbed by the portal vein, processed by the kidney, and excreted in the urine [45]. Bilirubin exists in concentrations between 10 and 80 mM in the gut of healthy individuals, with blood levels approximating 10 µM [46]. These levels are dependent on ethnicity and age, among other factors. Moreover, some pathological conditions (i.e., GS) are known for increasing the circulating concentrations around 17–80 µM [47].

Even if very high levels result in neurotoxic effects, bilirubin has been demonstrated to have different beneficial effects, such as antioxidant [48], anti-carcinogenesis [49], immunomodulatory [50], and anti-inflammatory effects on the CV system [51]. Moreover, it is involved in the regulation of several signaling pathways, thanks to its ability to bind receptors like aryl hydrocarbon receptor (AHR); CAR; Mas-related G protein-coupled receptor (MRGPR); peroxisome proliferator-activated receptor α (PPARα); peroxisome proliferator-activated receptor γ (PPARγ), thus acting as a hormone for energy homeostasis and lipid metabolism [39]. Indeed, serum-free bilirubin concentration has significant effects on morbidity and mortality in several pathologies, like diabetes, obesity, metabolic syndrome, and CVDs.

## 5. Bile Acids and Bilirubin in Oxidative Stress and Inflammation Related to Cardiovascular Disease

Despite BAs and bilirubin not being directly related in terms of their functions, they are both stored in the gallbladder in the form of bile and involved in the liver’s roles of digestion and waste elimination.

Indeed, elevated levels of BAs and bilirubin in the blood can be indicative of various health conditions, including liver diseases [28]. In the CV system, there is evidence to suggest that BAs and bilirubin have some opposite effects related to oxidative mechanisms, but this area of research is still under investigation [9,52,53]. In this paragraph, we report the main findings related to this topic, starting from BAs.

### 5.1. Bile Acids

As previously mentioned, BAs are produced in the liver and play a crucial role in the digestion and absorption of dietary fats in the small intestine. Because of that, they can indirectly contribute to anti-oxidation processes. Indeed, they help in the absorption of fat-soluble vitamins like A and E from the diet, which can then exert their antioxidant effects on the body. Alterations in the distribution, concentration, and composition of BAs (CA, CDCA, LCA, and DCA, in the free forms or conjugates) can lead to membrane damage caused by their detergent-like properties, which could be associated with oxygen-free radical production. Despite this, BAs are considered good oxidant scavengers at physiological concentrations found in bile and the intestine (10–80 mM). Indeed, they have been observed to protect both proteins (β-phycoerythrin) and lipids (linoleate and phosphatidylcholine) from peroxidation in vitro, suggesting that such protection could also take place in vivo [54]. Moreover, BAs can behave as antioxidants by trapping oxygen-free radicals within bile acid micelles depending on their structure and concentration [55]. The observed effects of the BAs on lipid peroxidation and oxidant scavenging result from the combination of the direct interception of peroxyl radicals and the variation in the composition of BA–fatty acid micelles at varying ratios of these components. LCA is the most hydrophobic BA and is not able to form micelles according to the lowest content of -OH moieties in the structure, so it cannot scavenge any radical species. LCA is also regarded as the most cytotoxic BA, whereas UDCA is the least toxic and the most hydrophilic [28].

Cardiac dysfunction is strongly associated with increased serum BA concentrations in patients with liver cirrhosis or other cholestatic diseases [53]. The biochemical mechanisms by which BAs exert these detrimental effects on myocardial function are still under investigation. Moreshwar Desai et al. have shown that an excess of BAs decreases fatty acid oxidation in cardiomyocytes and can cause heart dysfunction [56]; moreover, Mao et al. have recently demonstrated that heart failure-associated metabolic derangement results in cardiomyocyte cholesterol overload and accumulation of BA intermediates, which in turn triggers mitochondrial damage and inflammatory processes activation [57]. In patients with biliary obstruction, BAs have been observed to stimulate the generation of ROS from mitochondria, as well as promote their release from neutrophils and macrophages, thus causing oxidative damage to tubular cell membranes and the inflammatory consequences [58].

These results were also observed in heart mitochondria isolated from rats, where the most lipophilic BAs (LCA, DCA, and CDCA) altered mitochondrial respiration at systemic concentrations reached during cholestasis (levels up to several hundred micromolar) [8], thus determining an impairment also in intracellular ROS production.

Nowadays, there is a great interest in BSH-active bacteria that represent the main candidate for the prevention of hypercholesterolemia; indeed, numerous potential probiotics with high BSH activity significantly reduce circulating cholesterol levels (i.e., specific *Lactobacillus* and *Bifidobacteria* species such as *B. longum*, *L. salivarius*, *L. plantarum*, and *L. reuteri*) [24]. The interaction between BSH and FXR highlights the intricate crosstalk between the gut microbiota and the host. In line with these findings, numerous BSH-expressing probiotics have been shown to protect mice from weight gain and obesity, as well as to influence the BA pool by modulating FXR signaling [59]. FXR activation could lead to inhibition of the expression of the CYP7A1 gene by activating fibroblast growth factor 15 (FGF15) or SHP, contributing to reduced cholesterol levels and improving lipid profiles [36], thus markedly attenuating the development of atherosclerosis. Several studies have reported that FXR agonists activate the Nrf2 signaling, thus decreasing the production of ROS, malondialdehyde (MDA), and 8-Hydroxy-2-deoxyguanosine (8-OHdG). This occurs through elevated expression of catalase, glutathione-S-transferase, and superoxide dismutase (SOD) and has been shown to prevent cardiomyopathy in a diabetic mice model [36]. As previously mentioned, FXR has also been found in cardiomyocytes, vascular smooth muscle cells, and cardiac fibroblasts, clearly indicating that BAs may have an impact on the CV system [60].

Some studies have shown that pharmacological inhibition or genetic ablation of FXR significantly reduces myocardial apoptosis and rescues mitochondrial function in neonatal rat cardiac myocytes and fibroblasts while decreasing infarct size and improving cardiac function in ischaemic/reperfused myocardium in adult mice hearts [61,62]. In neonatal rat ventricular myocytes, its activation induces apoptosis due to the loss of mitochondrial membrane potential and cytochrome C release, as well as consequent caspase pathway activation [53]. Interestingly, FXR inhibition has been observed to reduce cardiotoxicity, decrease myocardial infarct size, and improve cardiac function in an ischemia-reperfusion model [53]. A growing body of research suggests that the hydrophilic UDCA may exert a protective role in CVD development since it acts as an FXR antagonist [63].

Those results suggest that the effect of BAs in FXR modulation is strongly “context-dependent” and should be analyzed more in-depth.

As previously mentioned, BAs act as signaling molecules, activating other receptors involved in metabolism and CV function other than FXR [14,28]. Activation of the nuclear receptors FXR, PXR, and VDR can influence lipid and glucose metabolism, inflammation, and blood pressure, which are key factors in CV health. As FXR, also nuclear PXR and VDR play an essential role in eliminating/mitigating BA-induced toxicity by downregulating the expression of CYP7A1, which is the liver rate-limiting enzyme for BA synthesis and is essential in maintaining lipid metabolism [14]. Interestingly, VDR in the t-tubules of cardiomyocytes [64] is involved in the regulation of intracellular calcium flow and contractile forces, thus being responsible for myocardial contraction [64]. Loss of VDR selectivity in cardiomyocytes leads to enlargement, hypertrophy, and systolic and diastolic dysfunction [64].

The activation of the G-protein coupled receptor TGR5 by BAs may have a protective effect on the CV system due to the improvement of vascular function. TGR5 is mainly expressed in aortic endothelial cells and plays an anti-atherosclerotic role by producing nitric oxide (NO) in a dose-dependent manner, inhibiting NF-κB activity, and regulating monocyte adhesion and the inflammatory response [65]. Moreover, the activation of TGR5 through the administration of taurine- CDCA and LCA has been observed in mouse cardiomyocytes to downregulate glycogen synthase kinase-3β (GSK3β) and upregulate protein kinase B (AKT), which are known to be associated with ROS production and cardiac hypertrophy [66]. Some studies also suggest that TGR5 activation by BAs is involved in the metabolic transformation of energy in cardiomyocytes [60]. S1PR2 is another BA-sensitive receptor found in vascular smooth muscle cells that is involved in NO signaling. Nevertheless, it works by inhibiting the synthase of inducible NO, thus reducing NO levels during a vascular injury [67]. 

Moreover, some researchers have suggested that certain BA derivatives (i.e., 7-cheto derivatives) and their metabolites may have indirect effects on oxidative stress and inflammation in the body [16]. However, this is still an emerging area of study, and the exact mechanisms are not fully understood yet. The principal biochemical mechanisms exerted by BAs in the CV system are reported in Figure 3.

### 5.2. Bilirubin

Bilirubin is generally considered a waste product with little or no physiological purpose and can be toxic if it accumulates. However, it has been recently recognized as a potent cardiac and vascular protective agent, suggesting its physiological importance [10]. Indeed, bilirubin reduces the risk of developing atherosclerosis and other CVDs through several inhibiting mechanisms, including low-density lipoprotein oxidation, vascular smooth muscle cell proliferation, and endothelial dysfunction [44,68,69]. In vivo experiments in apolipoprotein E-deficient (ApoE−/−) mice fed a Western-type (high fat) diet have shown that mildly elevated bilirubin levels significantly reduce plasma glucose, total cholesterol, low-density lipoprotein cholesterol, and the formation of atherosclerotic plaques [50]. Endothelial dysfunction is considered the early stage of atherosclerosis; it draws leukocytes to sites of injury/lipid accumulation, propagating vascular injury and encouraging vascular smooth muscle proliferation and neointima formation actions driven by the expression of adhesion molecules (i.e., intercellular adhesion molecule-1 (ICAM-1), vascular cell adhesion molecule-1 (VCAM-1), monocyte chemoattractant protein-1 (MCP-1) and oxidant-generating NADPH oxidases (NOXs).

It has been shown that bilirubin exerts a vascular endothelial protective effect on oxidant [8] and inflammatory [70] insults through several processes [9]. In endothelial cells, bilirubin has been observed to prevent tumor necrosis factor (TNF)-α-induced overexpression of adhesion molecules, such as E-selectin and VCAM-1 [70] and has shown to have an inhibitory effect on NOXs [71]. Oxidative stress derived from NOX activity promotes the expression of the inducible form of the enzyme HO-1, which in turn catalyzes the cleavage of heme to yield biliverdin, which is next reduced to the unconjugated “free” bilirubin, which in turn feeds back to quell oxidative stress [72] and has anti-inflammatory activities [73]. Animal studies in mice have shown that an excess of circulating bilirubin prevents the hypertensive effects of angiotensin II, together with the superoxide anion production generated by NOXs [74,75]. The potent effect of hyperbilirubinemia is not totally related to NOX inhibition, suggesting that bilirubin can act via both NOX-dependent and -independent mechanisms [75].

Bilirubin may also inhibit peroxynitrite (ONOO−) formation, a potent oxidant molecule resulting from the combination of NO and the superoxide anion. This inhibition occurs through the direct quenching of ONOO− and indirectly quenching/inhibiting superoxide anion, thus preventing the nitrosative injury [10]. This effect can protect the endothelium and, at least in part, determine an increase in NO bioavailability, which is associated with the relaxation and widening of blood vessels, even if a clear correlation between bilirubin and NO in humans is still under debate [76].

Bilirubin can also behave as a direct ROS scavenger, but its physiological intracellular content (in the nanomolar range) is too low to compete with other intracellular scavengers present in higher concentrations (e.g., glutathione and ascorbate) [77].

Interestingly, bilirubin levels increase after exercise, in line with increased skeletal muscle expression of HO-1 [78], suggesting that an increased bilirubin production might represent a stress-induced response to counteract detrimental ROS production and inflammatory mechanisms that occur during strong exercises. On the other hand, bilirubin acting as an ROS scavenger may interfere with stress-induced adaptation (i.e., pre-/postconditioning and exercise training).

It is important to remark that excessively high bilirubin levels, observed in some medical conditions like pancreatitis, GS, and some cancers, even conferring potent and versatile health protection, can be an indicator of liver or bile duct issues and adverse effects, including jaundice and gallstones, which themselves have CV implications [77].

In conclusion, although BAs and bilirubin appear to exert opposite effects on the CV system, it is worth noting that despite these interesting associations, definitive causation has yet to be established, and the relationship between them and CVD remains a subject of ongoing research. Additionally, the effects may vary from person to person and depend on several factors, including genetics. The principal biochemical mechanisms exerted by bilirubin in the CV system are reported in Figure 3.

## 6. Cardiovascular Diseases Associated with BAs/Bilirubin Deregulation: A Clinical Point of View

The data linking BAs to CVD risk in the general population are just a few; however, some epidemiological studies have identified intriguing associations between BAs and CV risk. In physiological conditions, fasting serum bile salt concentrations are typically below 5 μM and may rise to 5–10 mM in feeding conditions 1–2 h after meal intake [79]. Various studies found an association between CVDs and altered plasma bile salt concentrations and pool composition [79]. Indeed, one trial showed that in patients admitted to an Internal Medicine Unit due to chest pain and suspected coronary artery disease, lower amounts of total BAs, DCA, and LCA excretion were associated with stroke risk [80]. On the other side, serum BA levels were inversely related to the 3-month mortality of acute ischemic stroke patients, but a clear association with the severity of stroke or the incidence of complications has not been determined well so far [81]. In patients with moderate to severe chronic kidney disease, serum DCA (>58 ng/mL) was independently associated with a greater baseline coronary artery calcium volume score [82]. The effect of circulating BAs on CV health is more evident in specific diseases. For instance, high BA levels are associated with left ventricular mass and internal diameter in biliary atresia [83].

Primary biliary cholangitis is an autoimmune liver disease that mostly affects women, characterized by steatorrhea, weight loss, high plasma levels of total cholesterol, High-Density Lipoprotein-cholesterol (HDL-C), and lipoprotein X (Lp-X) [84]. In these patients, a higher prevalence of lower-limb atherosclerosis has been observed and related to microbiota composition modification [85].

Some drugs used to manage chronic liver disease with an impact on BA metabolism also have some effects on human plasma lipids levels. A meta-analysis of randomized clinical trials showed that UDCA significantly reduced total cholesterol (WMD: 29.86 mg/dL, 95% CI: −47.39, −12.33, *p* = 0.001) and LDL-cholesterol (WMD: −37.27 mg/dL, 95% CI: −54.16, −20.38, *p* < 0.001) concentrations without affecting triglycerides (TG) [86]. On the other side, obeticholic acid (OCA), which is a semi-synthetic BA analog (a highly selective agonist of FXR), is associated with a mild but significant reduction in HDL-C and apolipoprotein A-1 [87]. The impact of these observations on CV outcomes is yet to be clarified.

Currently, we have no definite evidence that pharmacologically modulating BA metabolism could be associated with reduced CV risk in humans, nor specific drugs in development targeting this mechanism of action for CV prevention [62]. However, monitoring bile salt levels and their pool composition in patients with CVDs could be an important aspect of understanding how these patients can optimally benefit from therapeutics targeting BAs [36].

Traditionally, the concentration of bilirubin in serum/plasma has been viewed as an indicator of hepatic disease. However, recent preclinical and pathophysiological data challenge this perception, revealing that elevated plasma bilirubin levels have a protective effect, while low levels may predispose individuals to CV and metabolic diseases [88], even if, from a clinical point of view, the relationship between bilirubin deregulation and CVDs is relatively controversial yet.

The GS, characterized by a benign, chronically mildly elevated bilirubin concentration in the blood, is mainly related to decreased hepatic bilirubin UDP-glucuronosyltransferase activity, decreased bilirubin clearance, and is associated with CVD protection [89]. Recently, 44,230 GS patients without atherosclerosis-related CVD, liver disease, diabetes, pregnancy status, or hypercholesterolemia (low-density lipoprotein ≥ 190 mg/dL) were compared with 44,230 matched overall healthy subjects. The study showed that individuals with GS consistently exhibited protective effects, and its magnitude increased with age while the mean bilirubin level remained constant [90]. However, in Mendelian randomization studies, genetically predicted bilirubin concentrations seem to not be associated with reduced CVD risk [91].

A very recent meta-analysis of 12 prospective studies (N = 368,567) concluded that the pooled risk ratio of CVD for the lowest vs. highest groups of bilirubin levels was 0.75 (95% CI: 0.58–0.97) with high heterogeneity (I^2^ = 87.5%, *p* < 0.001). Similar associations were observed separately for coronary heart disease and stroke. A “dose-response” meta-analysis of the same data showed a significant U-shaped relationship between circulating bilirubin and CVD (*p* < 0.01), with the lowest risk of CVD events observed in participants with a circulating bilirubin of 17–20 µmol/L. This evidence was clearer in men than in women [92].

This study confirmed a previous analysis of 25 prospective studies, including 316,375 subjects in primary prevention for myocardial infarction and with bilirubin in the normal range, in which increased bilirubin was associated with reduced risk of long-term (>3 years) first myocardial infarction by 22% (95% CI: 0.69–0.88). The dose-response analysis showed that the relative risk for first myocardial infarction decreased by 2.7% per each 2 μM increase of bilirubin (95% CI: 1.3%–4.1%, *p* < 0.001), with a cut-off value of 12.60 μM for relative risk >1.00. High blood bilirubin content also resulted in a reduced incidence of first and recurrent myocardial infarction by 36% (95% CI: 0.42–0.98) [93].

From a prognostic point of view, a recent meta-analysis of 43 studies, including 34,976 patients affected by coronary artery disease, showed that serum bilirubin was significantly higher in patients affected by myocardial infarction than in patients with non-myocardial infarction coronary artery diseases: 0.72 mg/dL [95% CI: 0.60, 0.83] in myocardial infarction patients; 0.65 mg/dL [95% CI: 0.60, 0.69] in non-MI CAD patients; and 0.66 mg/dL [95% CI: 0.56, 0.75] in healthy individuals. An increment of total bilirubin concentration was associated with adverse outcomes in myocardial infarction patients (OR: 1.08; 95% CI: 0.99, 1.18) but lower odds in non-myocardial infarction coronary artery disease patients (OR: 0.80; 95% CI: 0.73, 0.88). Compared to non-severe cases, total bilirubin levels were higher in patients with severe myocardial infarction (SMD 0.96; 95% CI: −0.10, 2.01; *p* = 0.074) but were lower in severe non-myocardial infarction coronary artery diseases patients (SMD −0.30; 95% CI: −0.56, −0.03; *p* = 0.02). Total bilirubin levels correlated positively with myocardial infarction severity (*r* = 0.41; 95% CI: 0.24, 0.59; *p* < 0.01) but negatively with non-myocardial infarction coronary artery disease severity (r = −0.17; 95% CI: −0.48, 0.14; *p* = 0.28). Female sex was inversely associated with increasing quantiles of bilirubin (meta-regression coefficient: −8.164; −14.531, −1.769; *p* = 0.016) in myocardial infarction patients. Therefore, the interpretation of the association between serum bilirubin and the prognosis of coronary artery disease is not easy or definitive [94].

Beyond coronary arteries, physiologically higher bilirubin levels have also been found to be inversely associated with diastolic blood pressure and new-onset hypertension, diagnosed using 24 h ambulatory blood pressure monitoring in perimenopausal women [95], and with a prevalence of carotid and femoral atherosclerosis in the general population [96].

Interestingly, regarding the sodium-glucose co-transporter 2 (SGLT2) inhibitors, antidiabetic drugs are able to significantly improve several CV risk factors and CV prognosis, also having a significant impact on bilirubin plasma levels. A recent meta-analysis of randomized clinical trials, including data from 1950 patients with type two diabetes, showed that SGLT2 inhibitors induced a significant increase in total plasma bilirubin (8.19% [0.79, 15.59], *p* < 0.01) and a significant decrease in serum alanine and aspartate aminotransferases, as well as in gamma–glutamyl transferase and liver steatosis (−3.39% [−6.01, −0.77], *p* < 0.01) comparing with placebo or other oral antidiabetic drugs [97]. Table 1 summarizes the most relevant clinical studies investigating the association between bile metabolism markers and cardiovascular outcomes in the general population and specific patient subgroups.

If a low bilirubin level (presumably < 10 μmol/L) is associated with a lower CVD risk, then we need to understand if its serum/plasma level should be considered as a risk factor or a risk marker of CVD. Consequently, we should identify the best cut-off to define a normal bilirubin level and, finally, if it makes sense, try to reduce it in both healthy individuals and patients [43].

Some healthy lifestyle suggestions are related to an increase in bilirubin levels. Indeed, if visceral obesity is associated with low bilirubin levels [98], it has been suggested that bilirubin levels linearly increase every 1% of body weight loss [99]. These observations are particularly of interest since bilirubin levels are associated with muscle mass, both in sarcopenia patients [100] and elite athletes [101]. The quality of diet also influences the bilirubin levels. A diet rich in fruit and vegetables [102] and whole grain carbohydrates [103] is associated with higher bilirubin levels, while high-fat diets [104] and diets rich in ultra-processed foods with low bilirubin levels [105]. Furthermore, aerobic physical activity is associated with higher bilirubin levels [46]. Of course, all these observations need more detailed evaluation because the impact of body weight optimization, the improvement of diet quality, and the increase of physical activity are per se associated with the prevention of CVD and recidivism, and it is yet not known how much bilirubin increase could be involved in the final positive preventive results.

Currently, there are no approved therapies specifically designed for plasma bilirubin level control. Potential approaches include (A) administering bilirubin in a water-based formulation orally or via injection, (B) using drugs to disrupt natural bilirubin conjugation in the liver, (C) exploring natural supplements affecting endogenous bilirubin metabolism, and (D) investigating the role of certain probiotics in altering bilirubin metabolism in the gut.

**Table 1 diseases-12-00103-t001:** Main studies investigating the association between bile metabolism markers and cardiovascular outcomes in the general population and specific patients’ subgroups.

Sample Size (N.)	Clinical Condition	Study Design	Marker	Outcomes	Main Results	Ref.
777	Acute ischemic stroke	Retrospective study	Bile acids	Stroke severity, in-hospital complication incidence, 3-month all-cause mortality	Bile acid levels were inversely associated with the 3-month mortality but not significantly associated with stroke severity or incidence of complications.	Huang et al. [81]
112	Moderate to severe Chronic Kidney Disease	Cohort analysis	Deoxycholic acid	Coronary artery calcification	Higher serum deoxycholic acid concentrations were independently associated with greater baseline coronary artery calcification.	Jovanovich et al. [82]
40 (Children)	Biliary atresia	Cohort analysis	Bile acids	Echocardiografic parameters	Bile acid concentrations >152 µmol/L were associated with an ∼8-fold increased odds of detecting abnormalities in left atrial and left ventricular geometry.	Virk et al. [83]
30 (Women)	Primary biliary cholangitis	Cohort analysis	Disease per se	Lower extremity arterial disease	Prevalence of lower extremity arterial disease in both NAFLD and control women (83.3% vs. 53.3% and 50%) and is associated with inflammatory markers and alterations in the gut-liver axis.	Ponziani et al. [85]
1370	Various liver diseases	Meta-analysis	Disease per se	LDL-cholesterol	Ursodeoxycholic acid therapy might be associated with significant total cholesterol lowering particularly in patients with primary biliary cirrhosis.	Simental-Mendia et al. [86]
44,230 (on a total population of 1,192,515)	Gilbert’s syndrome	Cross-sectional study	Disease per se	Atherosclerotic cardiovascular disease	Individuals with Gilbert’s syndrome consistently exhibited protective effect as they aged, and its magnitude increased with age.	Kartoun et al. [90]
463,060 participants in the UK Biobank (Replication in 429,209 subjects from the FinnGen)	Gilbert’s syndrome/Hyperbilirubinemia	Cohort study including observational, genetic, and Mendelian randomization analyses	Bilirubin; Disease per se	Cardiovascular diseases	Higher bilirubin concentrations (but not Gilbert genotype) had strong inverse associations with myocardial infarction, and cholesterol measures. However, genetic data suggest that bilirubin has no likely causal role in protection from cardiovascular disease.	Hamilton et al. [91]
368,567 participants	General population	Meta-analysis	Bilirubin	Cardiovascular diseases	There was a U-shaped dose-response relationship between bilirubin and cardiovascular disease, especially for men. The lowest risk of cardiovascular events was observed in participants with a bilirubin of 17–20 µmol/L in serum and/or plasma.	Zuo et al. [92]
316,375 subjects	General population	Meta-analysis	Bilirubin	Myocardial infarction	Higher bilirubin levels within a physiological range reduced the incidence of long-term first myocardial infarction, with a cut-off value of 12.60 μmol/L.	Yao et al. [93]
34,976 patients with CAD and 29,229 non-CAD individuals from general populations	General population	Meta-analysis	Bilirubin	Coronary artery disease (CAD)	Pooled serum total bilirubin levels were higher in myocardial infarction patients than in non-myocardial infarction CAD ones. Higher bilirubin levels were associated with greater odds of adverse outcomes in myocardial infarction patients, but lower odds in non- myocardial infarction CAD patients. Compared to non-severe cases, bilirubin levels were higher in patients with severe myocardial infarction, but lower in severe non- myocardial infarction CAD patients. Total bilirubin levels correlated positively with myocardial infarction severity, but negatively with non- myocardial infarction CAD severity.	Li et al. [94]
196 premenopausal women	General population	Cross-sectional study	Bilirubin	New-onset hypertension	In perimenopause, bilirubin was inversely associated with diastolic blood pressure and new-onset hypertension, diagnosed using 24 h ambulatory blood pressure monitoring.	He et al. [95]
4290 participants	General population	A Population-Based Cross-Sectional Study	Bilirubin	Femoral and Carotid Atherosclerosis	Increased serum bilirubin levels are inversely associated with the prevalence of carotid or femoral atherosclerosis. LDL and cholesterol may mediate these associations.	Su et al. [99]
100	Obese patients and healthy controls	Cross-sectional	Bilirubin	Cardiometabolic risk factors	Among cardio-metabolic risk factors, HDL-C and neutrophil-to-lymphocyte ratio are the most closely associated variables with bilirubin levels in obese adults.	El-Eshmawy et al. [98]

## 7. Conclusions and Future Perspectives

This study represents an attempt to review the literature regarding the role of BAs and bilirubin in oxidative stress and inflammation related to CV events.

To the best of our knowledge, this is the most complete overview of the literature regarding this topic. Despite this, it is important to remark that some relevant studies might be missing. In fact, because of the huge number of potentially relevant papers, we have used the AI-powered software ASReview (v1.5) to make the screening process more feasible and time-efficient. This has allowed us to retrieve more than one hundred relevant papers in a rather short period, avoiding the time-consuming activity of going through “noisy” or irrelevant papers during the screening of titles and abstracts. On the other hand, the use of ASReview with a quite flexible stopping rule could have caused some minor loss of information due to the failure to screen (and detect) potentially relevant papers. It is difficult to quantify the loss caused by this failure, but we guess that no more than 20 potentially relevant papers were missed. Moreover, based on the data available, it is reasonable to guess that at least 75% of them would have still been excluded during the full-text screening process, resulting in a loss of five relevant papers at most. Even keeping this limitation in mind, this work still provides an extensive overview of the evidence available for this field of study, which is relatively understudied to date.

A growing number of preclinical evidence supports the possible positive effect of bilirubin on CV health maintenance and CVD prevention beyond some direct protective effects in the heart and vessels. However, a comprehensive understanding of the interconnected intracellular events triggered by bilirubin in clinical trials remains elusive. Investigating whether modifying bilirubin levels through diet, exercise, and potential dietary supplementation can positively affect health, as indicated by population studies, is a crucial aspect.

In recent years, numerous attempts have been made to use bilirubin as a pharmacological agent. The methods have been different: HO-1 induction (by hemoglobin or myoglobin administration as enzyme substrate), administration of bilirubin precursors (i.e., biliverdin), inhibitors of the uridine diphosphate glucuronosyltransferase (and consequently of bilirubin conjugation), natural bilirubin or encapsulated in nanoparticles, or of synthetic analogs [106]. In particular, bilirubin, along with the induction of HMOX1, may emerge as novel therapeutics for conditions associated with metabolic and inflammatory dysfunction, such as CVD. Exposure to exogenous bilirubin or bilirubin-related drugs is safe in preclinical models overall. However, few of these therapeutic approaches have been tested in humans to date. For example, intravenous bilirubin has been tested in humans with a quite acceptable safety profile, but only for short periods [107]. Ongoing studies aimed at assessing the consequences of hypobilirubinemia [38] may offer valuable insights into understanding diseases like CV and fatty liver disease, where low bilirubin levels could be a contributing factor. Future trials will need to explore the possibility that increasing serum bilirubin levels are also associated with positive outcomes in terms of human CV health. Indeed, despite the development of several CV preventive drugs [108,109], the residual CV risk remains elevated in a relatively large number of patients [110,111].

Recently, evidence has accumulated indicating that BA metabolism disturbances and CVDs are closely related. When BA metabolism is disordered, a series of cardiac dysfunctions and CVDs may also be present [36]; this might be at least in part mediated by a dysregulated cholesterol metabolism. To the best of our knowledge, there are no well-documented and up-to-date clinical trials related to BAs as targets or biomarkers in CVDs, so further research is needed to fully elucidate their direct effects on oxidative stress and inflammation and to explore their potential therapeutic implications.

## Figures and Tables

**Figure 1 diseases-12-00103-f001:**
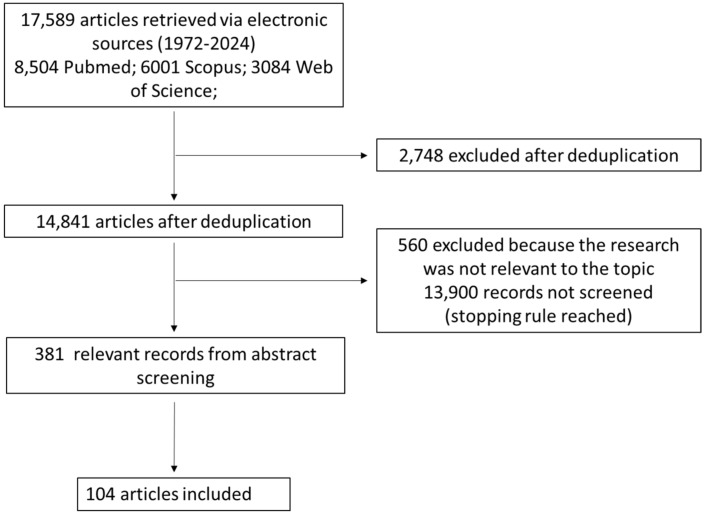
Flow chart of the bibliographic search strategy.

**Figure 2 diseases-12-00103-f002:**
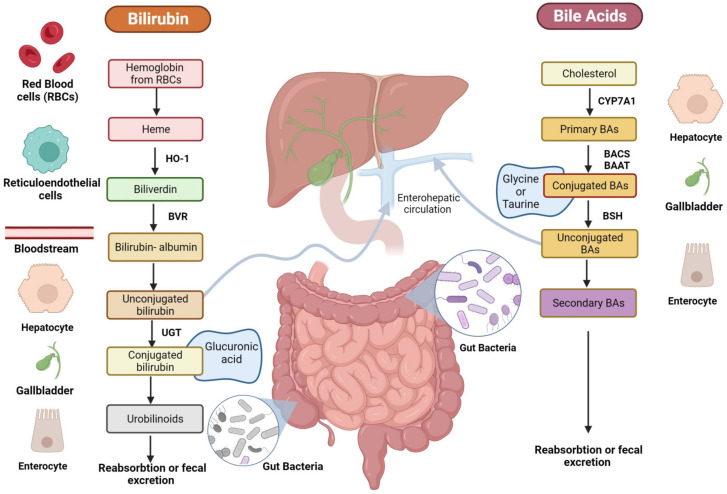
Schematic representation of BA and bilirubin pathways. The figure is original and created with BioRender.com (accessed on 10 December 2023).

**Figure 3 diseases-12-00103-f003:**
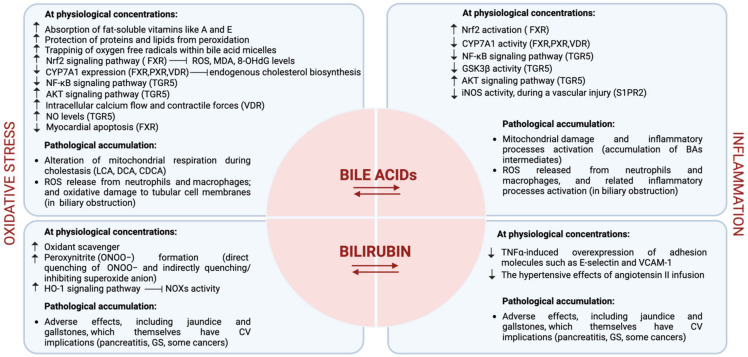
Representative figure of the biochemical mechanisms exerted by BAs and bilirubin in the CV system. The figure is original and created with BioRender.com.

## Data Availability

Not applicable.

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
