# Peer review of "Bile Acids and Bilirubin Role in Oxidative Stress and Inflammation in Cardiovascular Diseases"

_diseases, 2024, doi:10.3390/diseases12050103_

Round 1
Reviewer 1 Report
Comments and Suggestions for Authors
The manuscript diseases-2976778 entitled Bile Acids and Bilirubin Role in Oxidative Stress and Inflammation in Cardiovascular Diseases by Angela Punzo and coworkers, is a review work about the role of bile acids (BAs) and bilirubin on various physiological processes, including oxidative stress and inflammation. Clinical studies performed until now to assess the association between BAs/bilirubin and CVD were examined in depth to interpret the possible link between them better.
The work is interesting but too long.
The text is somehow misleading and some ideas are repeated several times despite their weakness.
A deep linguistic revision is required for another submission.
Minor point
· Line 41 and line 47: the connection between oxidative stress and inflammation in stated twice in few lines.
A revision of this section is recommended.
· Line 62: that bile acids are molecules is a consolidated concept. Please remove.
· Line 71-76: High levels of bilirubin (hyperbilirubinemia) may occur in certain conditions, such as jaun-71 dice or Gilbert Syndrome (GS). While excessively high levels of bilirubin can be a sign of 72 liver or other health issues, higher levels of bilirubin within the physiological range have 73 been associated with a reduced risk of coronary artery disease, peripheral artery disease, 74 and other manifestations of CVDs. Bilirubin's anti-inflammatory and antioxidant proper-75 ties are believed to contribute to its cardioprotective effects [10].
Better to reformulate as:
Increased serum concentration of bilirubin (hyperbilirubinemia, > 1.5 mg/dL) are typical of pathologic conditions such as jaundice or Gilbert Syndrome (GS) and is a sign of liver disease.
A bilirubin concentration at the higher degree within the physiological range was found to be associated with a reduced risk of coronary artery disease, peripheral artery disease and other manifestations of CVDs.
· Lines 106-183: this long section about bile acid metabolism report information which are available in several books and websites.
· Line 184-221: this long section about bilirubin metabolism report information available on several books and websites.
· Line 363: the evidence offered are of the 2014 and many things are changed.
Comments on the Quality of English Language
major linguistic revision
Author Response
The manuscript diseases-2976778 entitled Bile Acids and Bilirubin Role in Oxidative Stress and Inflammation in Cardiovascular Diseases by Angela Punzo and coworkers, is a review work about the role of bile acids (BAs) and bilirubin on various physiological processes, including oxidative stress and inflammation. Clinical studies performed until now to assess the association between BAs/bilirubin and CVD were examined in depth to interpret the possible link between them better.
· The work is interesting but too long. The text is somehow misleading and some ideas are repeated several times despite their weakness.
We thank the Reviewer for this comment; paragraphs 1, 2, 3, and 4 have been greatly shortened without compromising the understanding of the review. Moreover, we deeply checked the text to fix misleading and repeated concepts.
· A deep linguistic revision is required for another submission.
We thank the Reviewer for this suggestion; to improve the language and style used in this review, a complete and accurate revision was made by Dr. Matteo Perillo, a qualified biostatistician who contributed to this work also by implementing the literature search methodology.
Minor point
· Line 41 and line 47: the connection between oxidative stress and inflammation in stated twice in few lines. A revision of this section is recommended.
We thank the Reviewer for this comment; in the revised version of the manuscript, this issue has been addressed.
· Line 62: that bile acids are molecules is a consolidated concept. Please remove.
We thank the Reviewer for this comment; in the revised version of the manuscript, this issue has been addressed.
· Line 71-76: High levels of bilirubin (hyperbilirubinemia) may occur in certain conditions, such as jaun-71 dice or Gilbert Syndrome (GS). While excessively high levels of bilirubin can be a sign of 72 liver or other health issues, higher levels of bilirubin within the physiological range have 73 been associated with a reduced risk of coronary artery disease, peripheral artery disease, 74 and other manifestations of CVDs. Bilirubin's anti-
inflammatory and antioxidant proper-75 ties are believed to contribute to its cardioprotective effects [10].
Better to reformulate as:
Increased serum concentration of bilirubin (hyperbilirubinemia, > 1.5 mg/dL) are typical of pathologic conditions such as jaundice or Gilbert Syndrome (GS) and is a sign of liver disease.
A bilirubin concentration at the higher degree within the physiological range was found to be associated with a reduced risk of coronary artery disease, peripheral artery disease and other manifestations of CVDs.
We thank the Reviewer for this comment; in the revised version of the manuscript, this part has been rephrased.
· Lines 106-183: this long section about bile acid metabolism report information which are available in several books and websites.
In the revised version of the manuscript, this part has been reformulated and synthesized.
· Line 184-221: this long section about bilirubin metabolism report information available on several books and websites.
In the revised version of the manuscript, this part has been reformulated and synthesized.
· Line 363: the evidence offered are of the 2014 and many things are changed.
https://doi.org/10.1016/j.molmed.2023.01.007
10.1161/CIRCRESAHA.122.322418
https://doi.org/10.1016/j.jhep.2021.06.010
We thank the Reviewer for the suggestion. The first reference was already cited in the original version of the manuscript while we added the others. As we discussed in the new version of the manuscript (in the Discussion paragraph), the use of ASReview for the literature screening could have the intrinsic limitation of losing some relevant papers.
Reviewer 2 Report
Comments and Suggestions for Authors
In the review entitled "Bile Acids and Bilirubin Role in Oxidative Stress and Inflammation in Cardiovascular Diseases", the authors comprehensively and critically discussed the influence of bile acids and bilirubin on the development and progression of cardiovascular diseases, indicating future prospects in research aimed at medical applications. The manuscript is valuable and should be published in the journal Diseases after removing some minor shortcomings regarding text editing, listed below:
On the first three pages of the text, the distance between the lines (line spacing) is different than in the rest of the text. Please standardize the style throughout the manuscript as required by the editors.
At line 187 is … (Figure 1)[43] … , but should be … (Figure 1)[43] … . Comment: A space character is required before a literature reference.
At line 253 jest … The figure is original and created with BioRender.com. ... , but this text should be moved to the description of the figure to which it applies.
At line 303 is … plantarum, and L. reu- … , but should be … plantarum, and L. reu- … . Comment: The word "and" please write at normal style.
At line 312 is … 8-Hydroxy-2 -deoxyguanosine … , however, shouldn't it be without space ... 8-Hydroxy-2-deoxyguanosine ...?
At lines 373–374 is … [8]and inflammatory [77]insults … , but should be … [8] and inflammatory [77] insults … . Comment: Please insert a space character after the literature references.
At line 417 is … to 5–10 mM post-prandially with peak levels 1-2 hours … . Comment: Between the numbers there is sometimes a medium sign "–" and other times there is a short sign "-", but nowadays in scientific literature the medium sign is used. Please standardize throughout the manuscript.
At line 433 is … modification[93] … , but should be … modification [93] … . Comment: A space character is required before a literature reference. For example, at line 437 is … CI: -47.39, - 12.33, p= 0.001) and LDL-cholesterol (WMD: -37.27 mg/dL, 95% CI: -54.16, … . Comment: The mathematical subtraction sign “−” is desired throughout the manuscript. Please standardize the text at the body of the manuscript.
At line 513 is … p < 0.0.1) … , however, there are too many dots. Please check and correct.
On line 650 (ref. [30]) there is ... 1–20 ... , and it should be ... e1907271 ... . Comment: Please provide the article number, not the page range of the print version. Please check the entire source literature and remove any errors.
Author Response
In the review entitled "Bile Acids and Bilirubin Role in Oxidative Stress and Inflammation in Cardiovascular Diseases", the authors comprehensively and critically discussed the influence of bile acids and bilirubin on the development and progression of cardiovascular diseases, indicating future prospects in research aimed at medical applications. The manuscript is valuable and should be published in the journal Diseases after removing some minor shortcomings regarding text editing, listed below:
· On the first three pages of the text, the distance between the lines (line spacing) is different than in the rest of the text. Please standardize the style throughout the manuscript as required by the editors.
In the revised version of the manuscript, the line spacing has been standardized according to the journal guidelines.
At line 187 is … (Figure 1)[43] … , but should be … (Figure 1)[43] … .Comment:A space character is required before a literature reference.
· At line 253 jest … The figure is original and created with BioRender.com. ...,but this text should be moved to the description of the figure to which it applies. At line 303 is …plantarum, and L. reu-… , but should be …plantarum, and L. reu-… . Comment: The word "and" please write at normal style.At line 312 is … 8-Hydroxy-2 -deoxyguanosine … , however, shouldn't it be without space ...8-Hydroxy-2-deoxyguanosine...? At lines 373–374 is … [8]and inflammatory [77]insults … , but should be … [8] and inflammatory [77] insults … . Comment: Please insert a space character after the literature references.At line 417 is …to 5–10 mM post-prandially with peak levels 1-2 hours… . Comment: Between the numbers there is sometimes a medium sign "–" and other times there is a short sign "-", but nowadays in scientific literature the medium sign is used. Please standardize throughout the manuscript.At line 433 is …modification[93]… , but should be … modification [93]… . Comment: A space character is required before a literature reference. For example, at line 437 is …CI: -47.39, - 12.33, p= 0.001) and LDL-cholesterol (WMD: -37.27 mg/dL, 95% CI: -54.16,… . Comment: The mathematical subtraction sign “−” is desired throughout the manuscript. Please standardize the text at the body of the manuscript.At line 513 is … p < 0.0.1) … ,however, there are too many dots. Please check and correct.
We thank the Reviewer for his appreciation of our work. An accurate revision of the text was performed to correct errors and typos as his/her suggestions; abbreviations, spaces, and punctuation marks have been also checked.
· On line 650 (ref. [30]) there is ... 1–20 ... , and it should be ... e1907271 ... . Comment: Please provide the article number, not the page range of the print version. Please check the entire source literature and remove any errors.
We thank the Reviewer for this comment. In the revised version of the manuscript, this reference has been deleted as we reduced the length of the paragraph “Bile acids: Biochemistry and Physiology”. We also checked all the references, correcting some mistakes. The bibliography was generated using the software Mendeley and several references were changed in the revised version of the manuscript.
Reviewer 3 Report
Comments and Suggestions for Authors
Puzo and colleagues submitted a review of literature about the role of bile acids and bilirubin in cardiovascular diseases, with a focus on oxidative and inflammation mechanisms.
The authors explained the physiological and molecular basis of the implication of these two classes of compounds, describing the possible interaction with ROS and its role in cell signalling. Even if some protective action emerged from the literature, there are also scant information and conflicting results.
The authors tried to use the available pieces of information from preclinical and clinical studies, describing the evidence in humans within CV diseases and associated conditions.
Even if there is not a clear scenario, the authors described the aspects that need more in-depth studies to clarify some complex phenomena like the U-shaped relationship between CVD and bilirubin or data prom GS patients and the alleged protective role of this molecule.
The manuscript is well-written, fluent and detailed. There is a wide literature reference and the description of available data is impartial.
I have some suggestions to improve the interest of the reader:
- Even if this is a narrative review, I think it is better to provide a flow chart about the selection of sources
- Tables may increase the usability of the manuscript, so I advise to insert one or two tables about the selected studies
- A figure describing the signalling path of BA and bilirubin could improve the understanding of the reader
- The caption of Figure 1 is very redundant with the previous text. It can be omitted
- If I understand rightly, the stopping rules in the screening phase of sources may have limited some information, so I think the authors cannot say this review is comprehensive.
Author Response
Puzo and colleagues submitted a review of literature about the role of bile acids and bilirubin in cardiovascular diseases, with a focus on oxidative and inflammation mechanisms.
The authors explained the physiological and molecular basis of the implication of these two classes of compounds, describing the possible interaction with ROS and its role in cell signalling. Even if
some protective action emerged from the literature, there are also scant information and conflicting results.
The authors tried to use the available pieces of information from preclinical and clinical studies, describing the evidence in humans within CV diseases and associated conditions.
Even if there is not a clear scenario, the authors described the aspects that need more in-depth studies to clarify some complex phenomena like the U-shaped relationship between CVD and bilirubin or data prom GS patients and the alleged protective role of this molecule.
· The manuscript is well-written, fluent and detailed. There is a wide literature reference and the description of available data is impartial.
We are grateful to the Reviewer for the nice comment on our paper.
· I have some suggestions to improve the interest of the reader:
- Even if this is a narrative review, I think it is better to provide a flow chart about the selection of sources
We thank the Reviewer for this comment and, as suggested, we added a graphical flow chart of the study search (new Figure 1) in the paragraph “Search and Screening Strategy (Methods)”. We also listed the inclusion criteria of our literature search to better explain our study design, as follows:
“To be considered relevant, the papers had to comply with the following inclusion criteria:
-The paper discusses BAs and bilirubin synthesis and metabolism, physicochemical properties and their receptors and signaling pathways;
-The paper includes findings from human and animal studies (if relevant) with the support of preclinical data related to the role of BAs and bilirubin in CVDs.
- The non-availability of the full text was a criterion for exclusion.”
- Tables may increase the usability of the manuscript, so I advise inserting one or two tables about the selected studies
Following the Reviewer's comment, we inserted table 1 in the paragraph “Cardiovascular disease associated with BAs / Bilirubin deregulation: a clinical point of view”, which shows clinical studies that associate BAs or bilirubin with CV risk. The table is placed in a separate Word file and called ‘Table 1’.
- A figure describing the signalling path of BA and bilirubin could improve the understanding of the reader
We thank the Reviewer for this suggestion, which surely improves the understanding of the reader. We added a representative figure of the biochemical mechanisms exerted by BAs and bilirubin in CV system, named Figure 3, in the paragraph “Bile acids and Bilirubin in oxidative stress and inflammation related to cardiovascular disease” of the review.
- The caption of Figure 1 is very redundant with the previous text. It can be omitted
We agree with the Reviewer's suggestion, we deleted the description of BAs and bilirubin synthesis in the caption of the new Figure 2.
- If I understand rightly, the stopping rules in the screening phase of sources may have limited some information, so I think the authors cannot say this review is comprehensive.
We thank the Reviewer for this important comment; we agree that this search and screening strategy has some limitations due to the use of ASReview and the stopping rules we fixed. To better quantify the associated potential loss of information, we included an expert of ASReview as a co-author, Dr. Matteo Perillo, who helped us to better interpret the results of the screening procedure. A paragraph written by Dr. Perillo, describing the results of the search and screening procedure, has been included in the manuscript (subsection 2.2 in paragraph 2 named “Search and screening results”). Such results have been briefly discussed in the paragraph “Conclusions”.
In summary, while we are aware that some relevant studies might be missing due to the screening procedure followed, this review arguably represents the most complete revision of literature regarding this topic. Its comprehensiveness is still enhanced by the use of ASReview, which has been proven to contribute to the time efficiency and quality of the abstract screening for literature reviews.
Round 2
Reviewer 1 Report
Comments and Suggestions for Authors
The authors reviewed properly the full text of the manuscript.
Appreciable are the new figures.
English language is fine.
Author Response
Thank you very much for your nice appreciation
Reviewer 3 Report
Comments and Suggestions for Authors
I am sorry, I cannot find the Table 1!
All other aspects are satisfactory.
Author Response
Thank you very much for your nice appreciation. The Table 1 has now been included in the text.